# Neural network of social interaction observation in marmosets

**Justine C Cléry[1]\*, Yuki Hori[1], David J Schaeffer[1,2], Ravi S Menon[1,3], Stefan Everling[1,3]\***

[1]Centre for Functional and Metabolic Mapping, Robarts Research Institute, The University of Western Ontario, London, Canada; [2]University of Pittsburgh, Department of Neurobiology, Pittsburgh, United States; [3]Department of Physiology and Pharmacology, The University of Western Ontario, London, Canada

**Abstract** A crucial component of social cognition is to observe and understand the social interactions of other individuals. A promising nonhuman primate model for investigating the neural basis of social interaction observation is the common marmoset (*Callithrix jacchus*), a small New World primate that shares a rich social repertoire with humans. Here, we used functional magnetic resonance imaging acquired at 9.4 T to map the brain areas activated by social interaction observation in awake marmosets. We discovered a network of subcortical and cortical areas, predominately in the anterior lateral frontal and medial frontal cortex, that was specifically activated by social interaction observation. This network resembled that recently identified in Old World macaque monkeys. Our findings suggest that this network is largely conserved between New and Old World primates and support the use of marmosets for studying the neural basis of social cognition.

**\*For correspondence:**
jclery@uwo.ca (JCC);
severlin@uwo.ca (SE)

**Competing interests:** The authors declare that no competing interests exist.

## Introduction

Observing and interpreting social interactions of other individuals is crucial for our everyday life. We share this remarkable ability with other primates that can also readily recognize social interactions such as play, fight, and grooming of other group members (*Ambady et al., 2000*; *Ebenau et al., 2019*; *Spelke and Kinzler, 2007*). Monkeys use this information to infer social hierarchies, adjust their behavior, and find allies (*Bergman et al., 2003*; *Cheney et al., 1986*). In Old World macaque monkeys, *Sliwa and Freiwald, 2017* recently identified areas in parietal and frontal cortex that were exclusively activated by the observation of social interactions of other macaques using functional magnetic resonance imaging (fMRI). Interestingly, this network in macaques resembles the human default mode network (*Mars et al., 2012*) and the theory of mind network (*Gallagher and Frith, 2003*). As these networks share anatomical and functional characteristics, it suggests that the social interaction network may play a key role in these two human networks and be a conserved feature across primates (*Sliwa and Freiwald, 2017*).

Although macaque monkeys are widely used to study the neural basis of higher cognitive function, they may not be an ideal nonhuman primate model for studying the neural basis of social cognition as laboratory studies of social behavior are challenging due to their size and aggression toward conspecifics. Different species of macaques display varying type of prosociality but no cooperative breeding (*Balasubramaniam et al., 2018*; *Ballesta and Duhamel, 2015*; *Beisner et al., 2016*; *de Waal, 1986*; *Joly et al., 2017*; *McCowan et al., 2011*; *Thierry et al., 2000*) and are thus still an important model for studying some features and behaviors shared with humans (evolutionarily closer to humans than marmosets, larger size of societies, hierarchical societies, social behaviors including alliances, retaliation, redirected aggression). However, pairing and matching more than two macaque monkeys is often difficult under laboratory conditions. For these reasons, the common

marmoset (*Callithrix jacchus*), a small New World nonhuman primate species that lives in family groups and shares certain similarities with humans such as prosocial behavior, social cognition (*Burkart and Finkenwirth, 2015*; *Miller et al., 2016*), and cooperative breeding (*Burkart et al., 2009*; *Caselli et al., 2018*), has garnered interest as a powerful primate model for human social behavior (*Huang et al., 2020*; *Miller et al., 2016*; *Saito, 2015*; *Yokoyama et al., 2013*; *Yokoyama and Onoe, 2015*).

Here, we aimed to identify the neural circuits involved in social interaction observation in marmosets using ultra-high-field (9.4 T) fMRI in fully awake animals. Similar to Sliwa and Freiwald's study (2017), we acquired fMRI data from marmosets while they viewed videos of two marmosets interacting with each other (social condition) or of two marmosets engaged in independent goal-directed behavior (nonsocial condition).

## Results

In this study, we employed a block-design task that consisted of videos showing social interactions of two marmosets (social condition), two side-by-side videos showing a marmoset in each video interacting with its own environment (nonsocial condition), and phase-scrambled versions of these videos. In the social conditions, the actions performed in the videos included eating, playing, fighting, and grooming between two marmosets. In the nonsocial conditions, the actions performed in the videos included eating, exploring objects, looking around, and scratching. A more detailed description of these actions is provided in *Tables 1* and *2*. Three common marmosets were scanned, and eight runs per marmosets were used for group analysis (24 runs in total).

*Figure 1* shows group maps for four conditions (social, social scrambled, nonsocial, and nonsocial scrambled conditions) on fiducial brain surface (*Figure 1*, right hemisphere; see *Figure 1—figure supplement 1* for the left hemisphere and *Figure 1—figure supplement 2* for individual analyses, cluster-defining threshold. p<0.05, α = 0.05 for all contrasts). Both social (*Figure 1a*) and nonsocial (*Figure 1c*) videos activated a large network including visual (V1, V2, V3, V4, V4T, V6, V6A, 19M, MT), temporal (FST, TE1, TE2, TE3, TEO, IPa), parietal (MIP, VIP, LIP, AIP, PGM), and frontal areas (45, 47L, 46v, 10, 8av, 8ad, 8C, 6DR). The scrambled versions (*Figure 1b, d*) showed strong activation in visual (V1, V2, V3, V4, V4T, MT) and temporal areas (FST, TEO, IPa). Activations were also found in frontal cortex (47L, 45, 8av). The scrambled version of the nonsocial condition (*Figure 1d*) also showed activation in S2E, 1/2, 3a, 3b. Overall, the social and nonsocial conditions elicited a similar network, substantially stronger than those elicited by the scrambled versions.

To identify areas that were more active for the observation of social interactions, we compared the social with the nonsocial condition (*Figure 2*; see *Figure 2—figure supplement 1* for individual analyses, cluster-defining threshold p<0.05, α = 0.05 for all contrasts). This analysis revealed stronger activations in frontal, somatosensory, and cingulate cortex for the social condition. Significant differences were present in frontal areas 10, 9, 14c, 14r, 45, 46d, 47L, 8ad, 8b, 8av, 8C, 6av, 6bv, and 6DC. More posteriorly, somatosensory areas S2I, S2PV, S2PR, 1/2, 3a, and 3b, and motor areas 4ab

**Table 1.** Detailed description of the actions in each of the 'social interaction' condition for each order displayed, each clip within a block (two clips of 6 s) and each repetition (two repetitions) within the run.

| | Social interaction | |
|---|---|---|
| Order 1 | Playing (roll over the other and touching the head) | Grooming (one is lying down, the other one is sitting and grooming the other one's body) |
| | Grooming (one is lying down, the other is sitting and grooming the other's head) | Eating a biscuit (sharing food between a mother and her baby) |
| Order 2 | Playing in the forest (roll over the other and touching all body parts) | Playing in the forest (roll over the other and touching all body parts including playful biting) |
| | Grooming on a branch tree (one is lying down, the other one is sitting and grooming the other one's head) | Fighting and catching each other |
| Order 3 | Grooming on a branch tree (one is lying down, the other one is sitting and grooming the other one's head) | Fighting and catching each other |
| | Playing (roll over the other and touching the head) | Grooming (one is lying down, the other one is sitting and grooming the other one's body) |

**Table 2.** Detailed description of the actions in each of the 'nonsocial interaction' condition for each order displayed, each clip within a block (two clips of 6 s) and each repetition (two repetitions) within the run.

| | Nonsocial interaction | | | |
|---|---|---|---|---|
| Order 1 | Exploring a camera tripod | Eating (mostly chewing) | Eating banana peel | Scratching and exploring its environment |
| | Eating lettuce while lying on a nest bed | Eating on a tree branch | Eating on a tree branch | Eating while sitting on a table and looking at the camera |
| Order 2 | Eating banana and other fruits, frontal face view | Eating while sitting on a table and looking at the camera | Exploring and looking at its hand | Eating fruits profile view |
| | Frontal face view before eating sweet potatoes | Eating bananas and visually exploring | Looking backward, chasing insects | Looking backward before eating sweet potatoes |
| Order 3 | Eating lettuce while lying on in a nest bed | Eating on a tree branch | Eating on a tree branch | Eating while sitting on a table and looking at the camera |
| | Frontal face view before eating sweet potatoes | Eating bananas and visually exploring | Looking backward, chasing insects | Looking backward before eating sweet potatoes |

and ProM were more active for the social than nonsocial condition. We also observed activations in parietal areas (MIP, VIP), rostral temporal areas (STR, TPO, AuCPB, AuRPB), and in cingulate areas (25, 23b, and 31). The largest differences were present in motor/premotor cortex at the border of areas 4ab and 6DC, at the border of somatosensory area 1/2 and premotor areas 6av and 6bc, and in area 25 and orbitofrontal area 14. Differences between social and single interaction conditions are also found at the subcortical level in the SC, Tha, Amy, and CeB. Stronger activations for the nonsocial condition were mainly located in visual (V1, V2, V3, V4, V6), cingulate area 24b, and temporal areas (TF, TE2, TEO).

Overall numerous activations looked very similar in both hemispheres but are less strong in the left hemisphere (*Figure 2*; see *Figure 2—figure supplement 2* for temporal signal-to-noise ratio maps).

To better control for low-level visual features, we also reported the interaction effect [social vs. social scrambled] minus [nonsocial vs. nonsocial scrambled] (*Figure 3*; see *Figure 3—figure supplement 1* for individual analyses, cluster-defining threshold p<0.05, α = 0.05 for all contrasts). Overall, this analysis revealed most of the activations found by the previous contrast (*Figure 2*) but with less spread activations (*Figure 3*). Significant interaction effect was present in frontal areas 10, 9, 14c, 45, 46d, 47L, 8ad, 8b, 8C, 6bv, and 6DC, and more posteriorly, in somatosensory areas S2I, S2PR, 1/2, 3a, and 3b, motor area 4ab, and rostral temporal areas (STR, TPO, AuCPB). We still observed activations in parietal areas (VIP), in cingulate area 25, and at the subcortical level in the SC, Tha, Amy, and CeB. We did not observe any activation in area 8aV, indicating that the differences are not primarily related to saccades.

By extracting the time series from 12 regions of interest defined by the *Paxinos et al., 2011* atlas and based on the group activation maps, we assessed the differences of response magnitude between the four conditions (*Figure 4*). Some areas showed stronger activation for the social condition than for the nonsocial condition (paired t-tests, p<0.5; right area 25 p=0.0435; right area 10 p=0.0075; left area 10L p=0.0012; right area 14c p=0.0016; right area VIP p=0.0052; right area 6va p=0.027; right area 6vb p=0.021; right area 46d p=0.0186; left area 46d p=0.005; and right area 45 p=0.0291). These areas also showed stronger activations for the social condition than for its scrambled version (paired t-tests, p<0.05; right area 25 p=0.0355; right area 10 p=0.049; left area 10L p=0.011; right area 14c p=0.002; right area VIP p=0.0257; right area 6va p=0.038; right area 46d p=0.0074; and right area 45 p=2.8 $10^{-6}$) except for right area 6vb and left 46d.

Other areas did not show significant difference between social and nonsocial conditions but stronger activations for the social conditions than for its scrambled version (paired t-tests, p<0.05; left area 25 p=0.027; left area 6va p=0.012; right area 8ad p=0.007; left area 8ad p=0.014; right area 8av p=6.9 $10^{-6}$; left area 8av p=0.004; and left area 45 p=0.004).

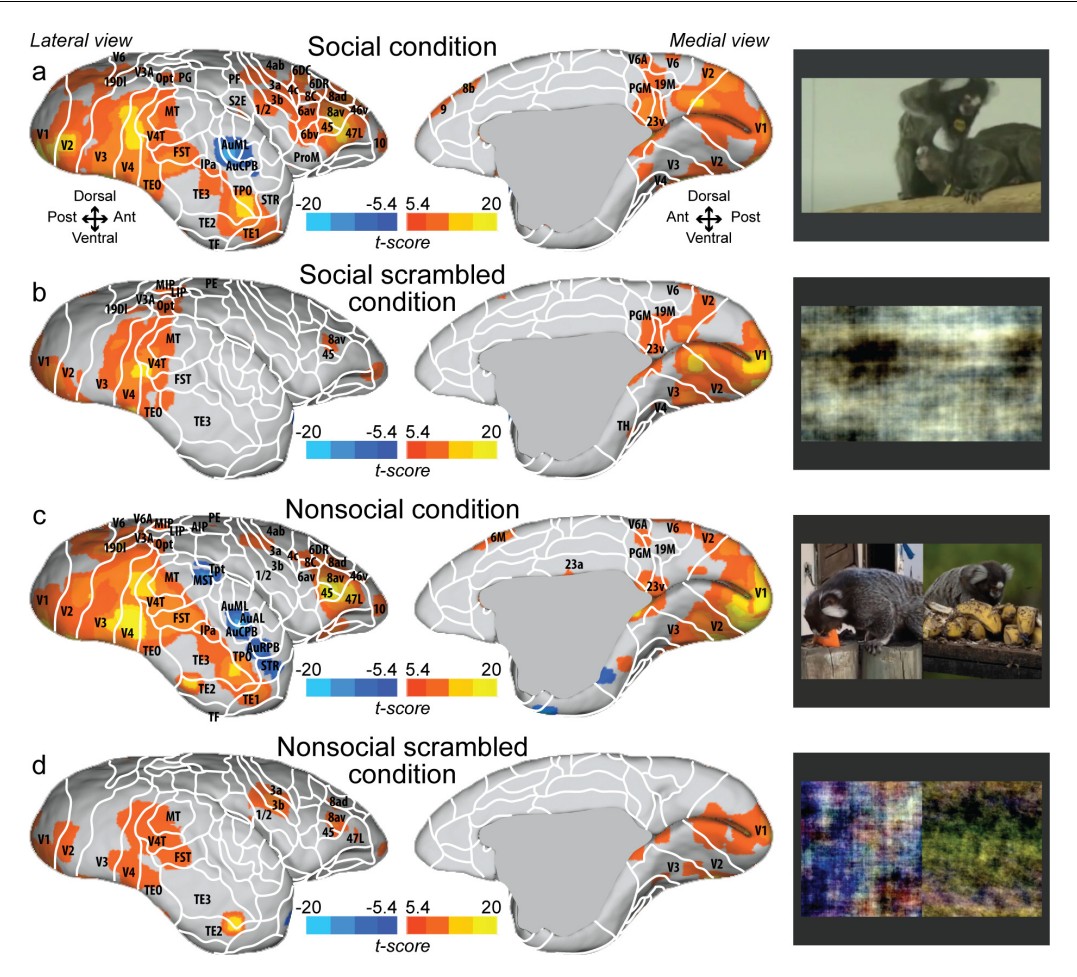

**Figure 1.** Group functional topology maps for social (**a**), social scrambled (**b**), nonsocial (**c**), and nonsocial scrambled conditions (**d**). Maps are displayed on the right fiducial brain surface (lateral and medial views). The white line delineates the regions based on the atlas from *Paxinos et al., 2011*. The regions associated with yellow/orange scale correspond to t-scores ≥ 5.4 (p<0.001, AFNI's 3dttest++, cluster-size correction from Monte Carlo simulation, α = 0.05). The regions associated with blue scale correspond to t-scores ≤ −5.4 (p<0.001, AFNI's 3dttest++, cluster-size correction from Monte Carlo simulation, α = 0.05). Group maps for the left hemisphere and individual maps for both hemispheres are shown in *Figure 1—figure supplement 1* and *Figure 1—figure supplement 2*, respectively.

The online version of this article includes the following figure supplement(s) for figure 1:

**Figure supplement 1.** Group functional topology maps for social (**a**), social scrambled (**b**), nonsocial (**c**), and nonsocial scrambled conditions (**d**).

**Figure supplement 2.** Individual functional topology maps for social (**a**), social scrambled (**b**), nonsocial (**c**), and nonsocial scrambled conditions (**d**) in monkeys 1–3.

## Discussion

In this study, we aimed to identify the neural circuitry underlying the observation of social interactions in New World marmosets. To that end, we employed a block-design task that consisted of videos showing social interactions of two marmosets (social condition), two side-by-side videos showing a marmoset in each video interacting with its own environment (eating, foraging) (nonsocial condition), and phase-scrambled versions of these videos. Both the social and nonsocial conditions showed strong occipitotemporal activation when compared to their scrambled versions, consistent with the location of face and body patches. In addition, these comparisons revealed activations in premotor and prefrontal areas. Like in macaques (*Sliwa and Freiwald, 2017*), the comparison of the social with the nonsocial condition revealed a network of mainly frontal cortical areas that is specifically activated by the observation of social interactions.

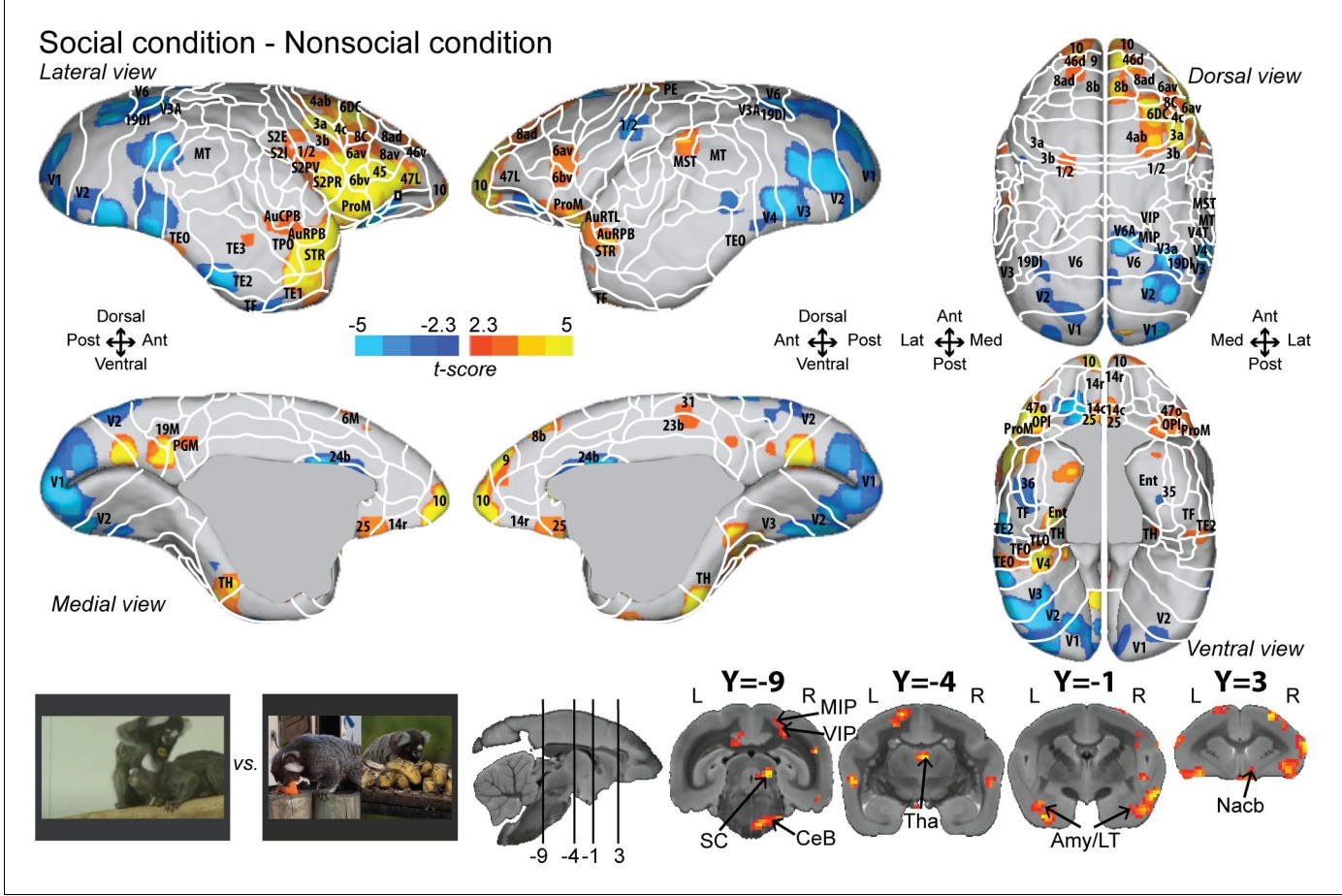

**Figure 2.** Group functional topology comparisons between social and nonsocial conditions. Groups maps are displayed on the right and left fiducial marmoset brain surface (lateral, medial, ventral, and dorsal views) and coronal slices. The white line delineates the regions based on the atlas from *Paxinos et al., 2011*. Y-coordinates are calculated with respect to the anterior commissure (in millimeters). The regions associated with yellow/orange scale correspond to t-scores $\geq$ 2.3 (p<0.05, AFNI's 3dttest++, cluster-size correction from Monte Carlo simulation, $\alpha$ = 0.05). The regions associated with blue scale correspond to t-scores $\leq$ –2.3 (p<0.05, AFNI's 3dttest++, cluster-size correction from Monte Carlo simulation, $\alpha$ = 0.05). Nomenclature for coronal slices is as follows: Amy/LT: amygdala/lateral temporal; CeB: cerebellum; MIP: medial intraparietal area; Nacb: nucleus accumbens; SC: superior colliculus; Tha: thalamus; VIP: ventral intraparietal area. Individual maps for both hemispheres are shown in *Figure 2—figure supplement 1*. The online version of this article includes the following figure supplement(s) for figure 2:

**Figure supplement 1.** Individual functional topology comparisons between social and nonsocial conditions.

**Figure supplement 2.** Temporal signal-to-noise ratio (tSNR) map for the group data.

The occipitotemporal activations in the intact vs. scrambled movies resemble face-selective patches previously found in marmoset fMRI studies using photos of marmoset faces and body parts (*Hung et al., 2015*) or videos of faces (*Schaeffer et al., 2020*). They include the occipital (V2/V3), anterior dorsal (rostral TE), middle dorsal (caudal TE), posterior dorsal (FST), and posterior ventral (V4/TEO) face patches. This was to be expected as marmoset faces were visible for most of the time during these conditions. We also confirmed a recent report of a frontal face patch located in the lateral frontal cortex (45/47L) in both the social and nonsocial condition (*Schaeffer et al., 2020*). These areas are thus recruited for face processing independent of social interactions.

Recently, *Sliwa and Freiwald, 2017* identified an exclusively social interaction brain network in macaques using fMRI. This network included predominately frontal and cingulate areas (14, 24c, 25, 32, 9, 10, 8B, 8D, 46D, 46V, 47L, 47O, 44, 6VR [F5], 6DC [F2], ProM). These overlap well with the frontal areas that we found here for a similar comparison in marmosets. In addition, we observed strong right hemisphere activation at the border of somatosensory area 1/2 and premotor areas 6av and 6bc and in the primary motor cortex 4ab in the marmoset.

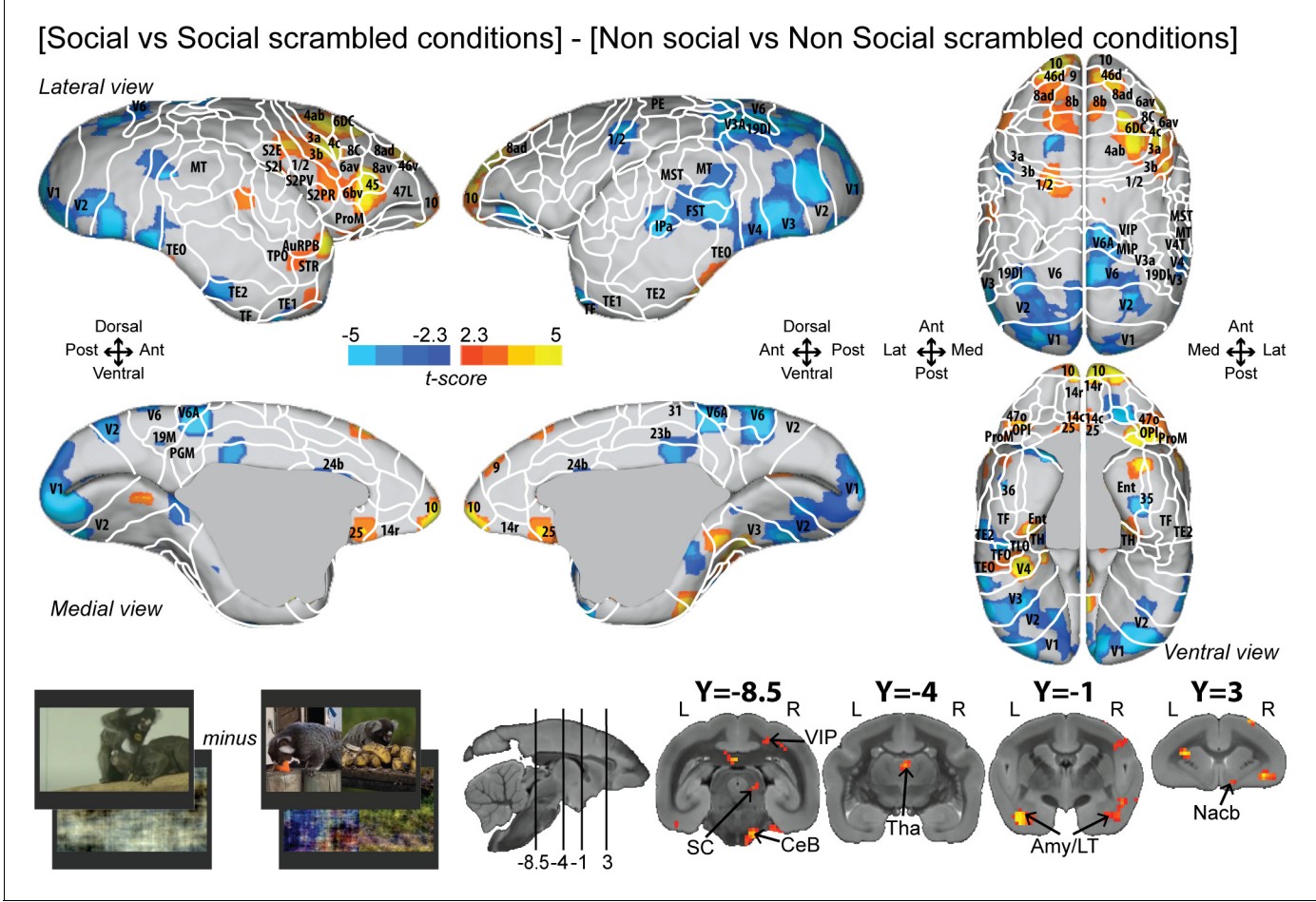

**Figure 3.** Group functional topology of the interaction effect [social vs. social scrambled] minus [nonsocial vs. nonsocial scrambled]. Groups maps are displayed on the right and left fiducial marmoset brain surface (lateral, medial, ventral, and dorsal views) and coronal slices. The white line delineates the regions based on the atlas from *Paxinos et al., 2011*. Y-coordinates are calculated with respect to the anterior commissure (in millimeters). The regions associated with yellow/orange scale correspond to t-scores ≥ 2.3 (p<0.05, AFNI's 3dttest++, cluster-size correction from Monte Carlo simulation, α = 0.05). The regions associated with blue scale correspond to t-scores ≤ −2.3 (p<0.05, AFNI's 3dttest++, cluster-size correction from Monte Carlo simulation, α = 0.05). Nomenclature for coronal slices is as follows: Amy/LT: amygdala/lateral temporal; CeB: cerebellum; Nacb: nucleus accumbens; SC: superior colliculus; Tha: thalamus; VIP: ventral intraparietal area. Individual maps for both hemispheres are shown in *Figure 3—figure supplement 1*.

The online version of this article includes the following figure supplement(s) for figure 3:

**Figure supplement 1.** Individual functional topology of the interaction effect [social vs. social scrambled] minus [nonsocial vs. nonsocial scrambled].

We also found activation of area PG (homologous with area 7a in macaque) in both social, social scrambled, and nonsocial condition but not during nonsocial scrambled version. In macaques (*Sliwa and Freiwald, 2017*), area PG/7a is strongly activated during social interaction observation but also showed, albeit weaker, activation during nonsocial interaction observation. In macaques, this area contains motor neurons coding goal-directed motor acts in macaques (*Rozzi et al., 2008*), which were present both in our social and nonsocial condition. The role of area PG is still poorly understood in marmosets, except that it seems to be a major functional hub of the marmoset brain (*Ghahremani et al., 2017*). Further studies need to be performed to fully unravel its function, in particular for social scrambled stimulation.

Most of the areas activated by the social vs. nonsocial comparison (and present in the interaction effect [social vs. social scrambled] minus [nonsocial vs. nonsocial scrambled]) are already known to form a strong structural network in marmosets and many have been implicated in various aspects of social cognition in humans and macaques. It should be noted that none of the oculomotor areas

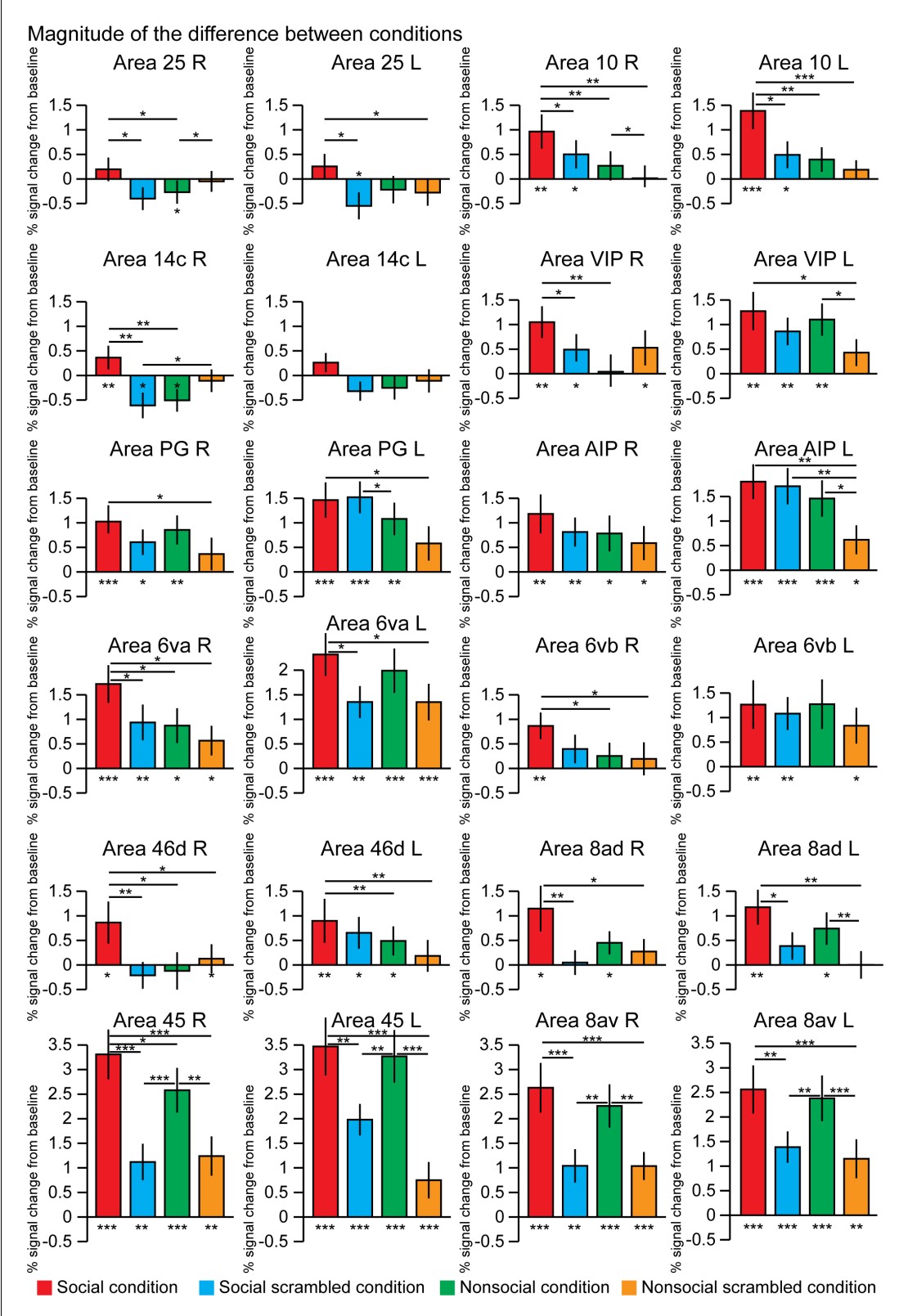

**Figure 4.** The magnitude of the difference between conditions within 12 regions of interests. These differences were calculated after extraction of the time series from these regions defined in the left and right hemisphere using the *Paxinos et al., 2011* atlas and based on the group activation maps. The differences from the baseline (below each bar graph) and conditions (horizontal bars) were assessed using paired t-test, *p<0.05, **p<0.01, ***p<0.001. The error bars correspond to the standard error.

were activated by this contrast, precluding any confounds with eye movements. Eye movements, and particularly visual saccades, mostly activate the lateral intraparietal area (LIP) and the frontal eye field in primates. Our group has mapped the saccade circuit in marmosets with multimodal approaches (electrophysiology and fMRI) (*Ghahremani et al., 2017*; *Johnston et al., 2019*; *Schaeffer et al., 2019b*; *Selvanayagam et al., 2019*). Together, this work demonstrates that saccades can be evoked by electrical stimulation of areas 45, 8aV, 8C, and 6DR. These regions are also activated in task-based fMRI with a visuomotor task and are functionally connected to the superior colliculus and other saccade-related regions as shown with RS-fMRI. In our contrast between the social and nonsocial conditions, only a small portion of these areas was activated and only in the right hemisphere. In addition, in the interaction effect contrast, no activation was present in area 8aV and only weak activations were observed in the dorsal part of 8aD and 6DR, suggesting that the activations we observed in our contrast are not directly related to oculomotor areas and so preclude most of the confound with eye movements. Furthermore, the PSC analysis of areas 8aV and 8aD showed no significant difference between social and nonsocial conditions. However, stimulation of area 45 evokes small saccades (*Selvanayagam et al., 2019*), and we observed here activation of the right area 45 in both social and nonsocial conditions as well as more activation for the social than nonsocial condition. The social condition may have elicited more small saccades related to the exploration of the facial expression and the interaction itself compared to the nonsocial interaction, which may involve more large explorations. This point raises some important limitations of our current setup. The animals were not rewarded for fixation or staying engaged during the task, and the quality of the video eye tracking did not allow us to confidently evaluate the animal's engagement/fixation across conditions and runs.

Area 10 is densely connected with areas 14 and 9, to the temporal pole (STR) and parabelt auditory cortex (*Buckner and Margulies, 2019*; *Burman et al., 2011*). Area 10 also receives projections from parahippocampal, ventral somatosensory, and premotor areas. It is a key node of a social cognition circuitry in humans (*Simmons et al., 2010*) and linked to areas integrating information about social attention, intentionality, and other's body actions in space in macaques (*Jellema et al., 2000*; *Nummenmaa and Calder, 2009*; *Oram and Perrett, 1996*).

Area 10 is also connected with area 25 in marmosets (*Burman et al., 2011*) and macaques (*Joyce and Barbas, 2018*). Connections with the amygdala, ventral striatum, hypothalamus, and periaqueductal gray, as well as with the anterior insula in macaques suggest a role of area 25 in emotion, visceromotor functions, and memory (*Alexander et al., 2020*; *Alexander et al., 2019*; *Joyce and Barbas, 2018*; *Öngür et al., 2003*). Our videos for the social condition showed marmosets playing or grooming each other, which may have triggered emotions and activated memories in the animals.

Tracer studies of area 14 in marmosets (*Burman and Rosa, 2009*), and some tracing data of the orbitofrontal cortex, encompassing area 14, have shown connections with amygdala, thalamus, and parahippocampal areas (*Roberts et al., 2007*). Area 14 seems to play a crucial role for appetitive value, reward processing, and decision making in humans (*Elliott et al., 2010*; *FitzGerald et al., 2009*) and in marmosets (*Roberts and Wallis, 2000*; *Roberts and Clarke, 2019*; *Stawicka et al., 2020*). Lesions in this area have been associated with social and emotional disturbance in humans (*Bechara et al., 1996*). Rostral area 14 also provides a major input to area 9 and send outputs to area 10 (*Burman et al., 2011*; *Burman et al., 2006*).

*Sliwa and Freiwald, 2017* also identified a parieto-premotor network involved in social interaction analysis. This network included the posterior part of area AIP and premotor area F5 that have been shown to contain mirror neurons in macaques (*Cléry et al., 2015*; *Cléry and Hamed, 2018*; *Rizzolatti et al., 1996*; *Rizzolatti and Fogassi, 2014*; *Rizzolatti and Matelli, 2003*; *Rizzolatti and Rozzi, 2018*). This literature, in addition to the study of *Sliwa and Freiwald, 2017*, suggests a role for this mirror system in recognizing and understanding other agent's intention and action. However, in this macaque study (*Sliwa and Freiwald, 2017*), this AIP/F5 network also responded to physical interactions between objects (not tested here), suggesting a broader role for this network. Here, we found that these areas were activated by the observation of nonsocial (i.e., 'agent-object interaction') and social interactions (i.e., 'agent-agent interaction') in marmosets. The mirror neuron system is still poorly understood in marmosets; however, neurons with characteristic 'mirror' properties have been identified in the superior temporal sulcus and in area 6V in marmosets (*Suzuki et al., 2015*).

Interestingly, the contrast between the social and nonsocial condition also revealed strong activations of parietal areas MIP and VIP in addition to premotor area 6V (corresponding to areas F4 and F5 in macaques, see *Matelli et al., 1985*). A parieto-frontal network related to VIP and F4 has been suggested to contribute to the construction of both the representation of one own's body and of the body of others in macaques (*Cléry et al., 2015*). Both areas VIP and F4 are strongly linked to peripersonal space representation, defense, avoidance behavior, and social cognition (*Brozzoli et al., 2013*; *Cléry et al., 2018*; *Cléry et al., 2017*; *Ishida et al., 2010*; for review, see *Cléry et al., 2015*; *Cléry and Hamed, 2018*). Our current study suggests that areas VIP and 6V also contribute to social interaction analysis in marmosets.

Even though the monkeys were simply watching other marmosets interacting with other marmosets or objects, we also found strong activations of somatosensory (1/2, 3a, 3b, SII) and motor areas (4ab, PGM) that were absent in macaque monkeys (*Sliwa and Freiwald, 2017*). Activations were localized in areas that are also activated by tactile stimulation of the face and arm (*Cléry et al., 2020*), body parts where tactile stimulation would often occur during social interactions (grooming, playing). Particularly SII (*Cléry et al., 2020*; *Huffman and Krubitzer, 2001*; *Krubitzer and Kaas, 1990*; *H-x et al., 2002*) was activated in both hemispheres. A role of SII in social interaction processing is consistent with its involvement in the observation of others' actions, as demonstrated both by human (*Keysers et al., 2004*; *Blakemore et al., 2005*; *Lee Masson et al., 2018*; for reviews, see *Keysers et al., 2010*; *Bretas et al., 2020*) and macaque studies (*Hihara et al., 2015*; *Raos et al., 2014*).

Our data showed stronger activations in the right hemisphere compared to the left hemisphere, even if the overall pattern looked similar. The temporal signal-to-noise ratio (tSNR) map showed a slightly lower tSNR in the left hemisphere (*Figure 2—figure supplement 2*), which may have led to the difference of activation we observed. However, this difference may also be due to a real lateralization effect during social behaviors. Indeed, it has been shown that the right hemisphere has a relative advantage over the left hemisphere, particularly in identifying social stimuli, understanding the intentions of others, social relationships, and social interactions in humans (see review, *Hecht, 2014*). Consequently, our data may reflect a similar lateralization for social processing in marmosets.

An interesting finding with our current study is the deactivation observed in both the social and nonsocial condition that were not present during the scrambled versions. This deactivation may have been triggered by an expectation of hearing vocalizations that were not met, but the actions during nonsocial interaction were mainly associated with eating so there is not necessarily vocalization associated with this kind of action. However, the fact that we observed this deactivation in both interaction conditions and not in the scrambled conditions may be the result of the marmosets' interest and engagement during the videos. Looking at conspecifics interacting with each other or with their environment is more entertaining than blurred videos. Further studies need to be performed to test a more complex paradigm such as videos showing social interactions with or without vocalizations.

In summary, we have identified the neural circuitry for social interaction observation in marmosets that resembles the one identified in Old World macaque monkeys (*Sliwa and Freiwald, 2017*). As Old and New World primates separated ~40 million years ago, this suggests that neural circuits involved in social interaction observation are conserved between macaques and marmosets or may be the result of convergent evolution (they may have underwent similarities in selection pressure but also show differences in their social behavior; *Preuss, 2019*). Social interaction observation recruited multiple brain networks, at both the cortical and subcortical levels, likely involved in emotion and affective processing, memory, peripersonal space representation, and the mirror neuron system. Overall, the identification of a preserved cortical and subcortical circuitry involved in social interaction observation in marmosets provides the map for studying the neural processes underlying primate social cognition.

## Materials and methods

All experimental methods described were performed in accordance with the guidelines of the Canadian Council on Animal Care policy on the care and use of experimental animals and an ethics protocol #2017-114 approved by the Animal Care Committee of the University of Western Ontario.

## Subjects and experimental setup

Three common marmosets (*C. jacchus*) completed this study (one female, two males). Their ages and weights at the time of the experiment were 30 months and 363 g (M1), 38 months and 410 g (M2), and 35 months and 310 g (M3), respectively. The animals were prepared for awake fMRI experiments by implanting an MRI-compatible head restraint/recording chamber (for details, see *Johnston et al., 2018*). Each animal was initially habituated to the MRI environment over the course of 3 weeks. This included the acclimatization to the animal holder (tube closed by neck and tail plates), the head-fixation system, and periodic sounds produced by the MRI sequences (for details, see *Schaeffer et al., 2019a*). The behavior of the animals was monitored throughout the training to assess the tolerance and well-being of the marmosets (*Silva et al., 2011*). Their progression was based on their performance on the behavioral scale defined by *Silva et al., 2011*.

For MRI acquisition, the animal was placed in the sphinx position in an animal holder consisting of an MRI-compatible restraint system (*Schaeffer et al., 2019a*). The animal was first restrained using neck and tail plates, then the head was restrained by the fixation of the head chamber to the five-channel receive coil. An MRI-compatible camera (model 12M-i, 60 Hz sampling rate, MRC Systems GmbH, Heidelberg, Germany) was positioned in front of the animal holder to allow monitoring of the animals during the acquisition sessions by a veterinary technician. However, this camera and the setup did not allow a proper recording of eye movements as the pupil signal was frequently lost. In the MR scanner, the animal faced a translucent plastic screen placed at the front of the bore (119 cm from the animal's head) where visual stimuli were displayed via back-projection after being reflected off of a first-surface mirror. Images were projected with a SONY VPL-FE40 projector. The maximum visual angle from the center to the side of the screen was 6.5°. Stimuli were presented via Keynote (version 10.1, Apple Incorporated, CA) and launched in synchronization with MRI TTL pulse triggers via a custom-written program running on a Raspberry Pi (model 3 B+, Raspberry Pi Foundation, Cambridge, UK).

## Task and stimuli

Four task conditions were presented: social condition videos where two marmosets directly interacted with each other (playing, grooming …) and nonsocial videos where two marmosets interacted with their own environment (eating, foraging) in two separate videos but displayed at the same time (*Figure 1a, c*, right panel). We chose real-world videos to maximize cognitive engagement (*Finn and Bandettini, 2020*; *Meer et al., 2020*). Videos were edited into 12 s clips using custom video-editing software (iMovie, Apple Incorporated). Two video clips were displayed within a block (6 s each), and 20 different videos clips were used in total. The videos within the same run were different between the two repetitions of the block condition but played following the same order. *Tables 1* and *2* describe which actions were performed for each social (*Table 1*) and nonsocial (*Table 2*) blocks. Scrambled versions of social interaction and nonsocial interaction videos (*Figure 1b, d*, right panel) were created by random rotation of the phase information using a custom program (MATLAB, The MathWorks, Matick, MA). To preserve motion and luminance components, the same random rotation matrix was used for each frame in the scrambled conditions. Videos were presented in the center of the screen (7° height × 13° width) during acquisition sequences.

These four pseudo-randomized conditions were each displayed twice (social condition, nonsocial condition, and scrambled versions of each) following a block design (*Figure 5*). These eight task condition blocks (12 s each) were separated by baseline blocks (18 s each) where a 0.36° circular black cue was displayed in the center of a gray background. Three different stimulus sets were used and counterbalanced within the same animal and between animals. A sequence lasted 253 s (4.3 min). As motion and luminance varied across videos, mainly due to where the action took place, we counterbalanced these differences across the selected videos, conditions, and order of presentation to avoid any link with the condition block itself. No reward has been provided to the monkeys during the scan session.

## Scanning

We performed data acquisition using a 9.4 T, 31 cm horizontal-bore magnet (Varian/Agilent, Yarnton, UK) and Bruker BioSpec Avance III HD console with the software package Paravision-6 (Bruker BioSpin Corp, Billerica, MA) at the Centre for Functional and Metabolic Mapping at the University of

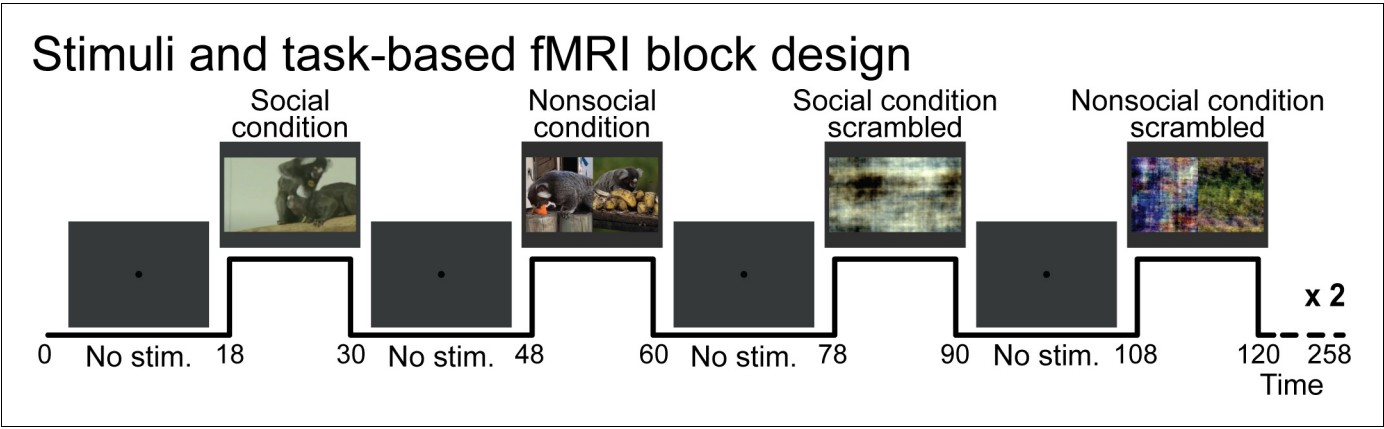

**Figure 5.** Stimuli and task-based functional magnetic resonance imaging block design. Social condition videos, nonsocial condition videos, and their scrambled versions were displayed in a screen during task conditions blocks (12 s each) and separated by baseline blocks where a central dot was displayed in the center of the screen (18 s each). A gray background was displayed behind the dot and videos through the sequence.

Western Ontario. We used an in-house, custom-built integrated receive coil with five channels (*Schaeffer et al., 2019a*) paired with a custom-built high-performance 15-cm-diameter gradient coil with 400 mT/m maximum gradient strength (xMR, London, Canada; *Peterson et al., 2018*). We used an in-house quadrature birdcage coil (12 cm inner diameter) for the transmit coil.

For each animal, a T2-weighted structural image was acquired, to allow an anatomical registration, with the following parameters: repetition time (TR) = 5500 ms; echo time (TE) = 53 ms; field of view (FOV) = 51.2 × 51.2 mm; voxel size = 0.133 × 0.133 × 0.5 mm; number of slices = 42 (axial); bandwidth = 50 kHz; GRAPPA acceleration factor = 2. For functional imaging, gradient-echo-based, single-shot echo-planar images covering the whole brain were acquired over multiple sessions (TR = 1500 ms; TE = 15 ms; flip angle = 40°; FOV = 64 × 64 mm; matrix size = 128 × 128; voxel size = 0.5 mm isotropic; number of slices = 42 [axial]; bandwidth = 500 kHz; GRAPPA acceleration factor = 2 [anterior-posterior]).

## Analysis

Monkey 1 performed 8 runs in four sessions, monkey 2 performed 11 runs in five sessions, and Monkey 3 performed 9 runs in four sessions. We selected eight runs per monkeys to have the same number of runs across subjects for the group analysis (the selection was based on the best engagement of the animal as assessed by visual monitoring during the sessions). Time series were preprocessed using AFNI (*Cox, 1996*), FSL (*Smith et al., 2004*), and ANTS software (Advanced Normalization Tools, *Avants et al., 2011*). Raw functional images were converted to NifTI format using dcm2niix (*Li et al., 2016*) and reoriented from the sphinx position using FSL. The images were then despiked (AFNI's 3dDespike) and volume registered to the middle volume of each time series (AFNI's 3dvolreg). The motion parameters from volume registration were stored for later use with nuisance regression. Images were smoothed by a 1.5 mm full-width at half-maximum Gaussian kernel to reduce noise (AFNI's 3dmerge). An average functional image was then calculated for each session and linearly registered (FSL's FLIRT) to each animal's T2-weighted image – the 4D time-series data was carried over using this transformation matrix. Anatomical images were manually skull-stripped, and this mask was applied to the functional images in anatomical space. The T2-weighted images were then nonlinearly registered to the NIH marmoset brain atlas (*Liu et al., 2018*) using Advanced Normalization Tools and the resultant transformation matrices stored for later transformation (see below). The olfactory bulb was manually removed from the T2-weighted images of each animal prior to registration as it was not included in the template image.

The task timing was convolved to the hemodynamic response (using AFNI's 'BLOCK' convolution), and a regressor was generated for each condition to be used in a regression analysis (AFNI's 3dDeconvolve) for each run. All four conditions were entered into the same model, along with polynomial (N = 5) detrending regressors, bandpass regressors, and the motion parameters derived from the volume registration described above (these nuisance regressors correspond to the baseline

variance for the statistical analysis model). The resultant regression coefficient maps were then registered to template space using the transformation matrices described above. The T-value maps for each monkey were then compared at the individual level via paired-sample t-tests and fixed effect analyses across the eight runs for each marmoset (AFNI's 3dttest++) and at the group level via a three-way analysis of variance and mixed effect analysis (AFNI's 3danova3) using eight runs per marmosets (for a total of 24 runs). To protect against false positives, a cluster-size correction derived from Monte Carlo simulations was applied to the resultant t-test maps for $\alpha = 0.05$ (using AFNI's 3dClustsim).

The contrast between the social and nonsocial condition as well as the interaction effect [social vs. social scrambled] minus [nonsocial vs. nonsocial scrambled] was compared to previous fMRI and electrophysiological studies performed in marmosets (*Ghahremani et al., 2017*; *Johnston et al., 2019*; *Schaeffer et al., 2019b*; *Selvanayagam et al., 2019*) to check if the observed activations may be related to eye movements.

The resultant T-value maps are displayed on coronal sections or fiducial maps obtained with Caret (*Van Essen et al., 2001*; http://www.nitrc.org/projects/caret/) using the NIH marmoset brain template (*Liu et al., 2018*). Coordinates are presented with respect to the anterior commissure. The labeling refers to the histology-based atlas of *Paxinos et al., 2011* for the cortical regions and to the atlas of *Liu et al., 2018* for the subcortical regions (*Figure 1*, *Figure 1—figure supplement 1*, *Figure 2*, *Figure 3*).

Twelve regions of interest, areas 25, 10, 14c, VIP, PG, AIP, 6va, 6vb, 46d, 45, 8av and 8ad, were defined based on the *Paxinos et al., 2011* atlas to evaluate the difference in response magnitude between conditions. First, masks of these regions were generated using AFNI's 3dcalc in the right or left hemispheres of the 'NIH_Marmoset_atlas_Paxinos_cortex'. Then, time series were extracted for each condition from the resultant regression coefficient maps using AFNI's 3dmaskave. The mean and standard errors were computed and paired t-tests were calculated using MATLAB (version 2018a) after the data were tested for normality (Kolmogorov–Smirnov test).

## Acknowledgements

Support was provided by the Canadian Institutes of Health Research (FRN 148365) and the Canada First Research Excellence Fund to BrainsCAN. We thank Miranda Bellyou, Cheryl Vander Tuin, and Hannah Pettypiece for animal preparation and care and Dr. Alex Li for scanning assistance.

## Additional information

### Funding

| Funder | Grant reference number | Author |
| --- | --- | --- |
| Canadian Institutes of Health Research | FRN 148365 | Stefan Everling |
| Canada First Research Excellence Fund | BrainsCAN | Stefan Everling |

The funders had no role in study design, data collection and interpretation, or the decision to submit the work for publication.

### Author contributions

Justine C Cléry, Data curation, Formal analysis, Investigation, Visualization, Methodology, Writing - original draft, Writing - review and editing; Yuki Hori, Data curation, Investigation, Methodology, Writing - review and editing; David J Schaeffer, Investigation, Methodology, Writing - review and editing; Ravi S Menon, Resources, Validation, Writing - review and editing; Stefan Everling, Conceptualization, Supervision, Funding acquisition, Validation, Investigation, Methodology, Project administration, Writing - review and editing

### Author ORCIDs

Justine C Cléry (iD) https://orcid.org/0000-0003-1020-1845

## Ethics

Animal experimentation: All experimental methods described were performed in accordance with the guidelines of the Canadian Council on Animal Care policy on the care and use of experimental animals and an ethics protocol #2017-114 approved by the Animal Care Committee of the University of Western Ontario. Animals were monitoring during the acquisition sessions by a veterinary technician.

## Decision letter and Author response

Decision letter https://doi.org/10.7554/eLife.65012.sa1
Author response https://doi.org/10.7554/eLife.65012.sa2

## Additional files

### Supplementary files

• Transparent reporting form

### Data availability

The datasets generated during this study are available at https://github.com/JClery/Social_interaction_paper and a copy archived at https://archive.softwareheritage.org/swh:1:rev:c9381727d3448d97d4f4263c1bb34c68ca296968/.

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
