## [Decision Letter]

**Acceptance summary:**

This marmoset fMRI study will be important for researchers interested in social cognition and non-human primate models. The results describe a brain network activated when marmoset monkeys observe social and non-social interactions. This network is similar to regions previously shown to be involved in processing social information in humans and macaques.

**Decision letter after peer review:**

Thank you for submitting your article "Social interaction observation network in marmosets" for consideration by *eLife*. Your article has been reviewed by 3 peer reviewers, one of whom is a member of our Board of Reviewing Editors, and the evaluation has been overseen by Tamar Makin as the Senior Editor. The reviewers have opted to remain anonymous.

The reviewers have discussed the reviews with one another and the Reviewing Editor has drafted this decision to help you prepare a revised submission.

Summary:

This fMRI study maps brain areas activated when marmoset monkeys observe social interactions. During fMRI data acquisition, animals viewed short videos of social interactions, non-social interactions, scrambled social interactions, or scrambled non-social interactions. The authors report that a distributed set of brain regions is involved in processing social information. Importantly the identified regions are similar to regions previously shown to be involved in processing social information in humans and macaques.

All reviewers agreed that your manuscript addresses and important question and that your findings will fill an important gap in the literature. However, reviewers also identified several issues that need to be addressed in a revision.

Essential revisions:

1. It would be important to more clearly describe the statistical group-level model (fixed-effect, mixed effect, random effect?). Most importantly, it was not clear whether and how the group model accounted for the nested structure of technical (8 runs) and biological replicates (3 animals). Specifically, it seems the 24 runs were treated as independent observations (given they are not independent, they should not be treated as such). Related to this, it would be important to also show results for each of the individual 3 subjects.

2. There was a general concern that the description of stimuli and the quantification of animal behavior was insufficient. This concern could be overcome by providing additional information and analyses, as listed below.

2.1. L75/L158/L297 Please provide a more detailed description of the actions in each of the 'social' and 'nonsocial' videos in the Results and Methods sections, and provide the test videos as supplementary material.

2.2. L 93 "AFNI's 3dttest++" Please describe in the Methods section the kind of test that is coded by this AFNI function (fixed effect? Mixed effect analysis? corrected for multiple comparison?).

2.3. Figure 1: What is the baseline used for comparison in the GLM and tests? Were all the baseline times concatenated? A baseline in the middle or at the end or at the beginning of each run?

2.4. L192 "It should be noted that none of the oculomotor areas were activated by this contrast, precluding any confound with eye movements". This is an important information appearing only in the Discussion. Please describe in the Methods section how it was assessed, and in the Results section provide more details on this result.

2.5. L279-280 "this camera and setup did not allow a proper recording of eye movements": Did it allow to monitor other interesting information: e.g. how long eyes were open during each condition, and/or to monitor blinks?

2.6. L296-306: Was there one video per block? Is there a difference in motion or luminance between conditions?

2.7. L307-312: Are conditions presented in reverse order the second time during the run? Which baseline is used for contrasts presented in figure 1? Where the animals rewarded with juice during the scan? If so, for what type of behavior (fixation during baseline? looking at the videos, looking at the screen? Staying engaged?).

2.8. Additional information should be given regarding time courses of regions highlighted on maps and in the discussion, ie PG, PMv, AIP, 8A/45.

2.9. How many video clips were used in total? How many runs/sessions were collected?

3. Contrasts in Figure 2 and 3 are shown at p<0.05, uncorrected. Given the large number of voxels, whole brain tests should be corrected for multiple comparisons. Alternatively, the should be displayed at a more conservative uncorrected threshold (e.g., p<0.001 as for) as in Figure 1.

4. The authors isolate areas involved in observing social interactions by comparing responses to social and non-social interactions. To better control for low level visual features, please report the interaction effect ([social minus social scrambled] minus [nonsocial minus nonsocial scrambled]).

---

## [Author Response]

Essential revisions:1. It would be important to more clearly describe the statistical group-level model (fixed-effect, mixed effect, random effect?). Most importantly, it was not clear whether and how the group model accounted for the nested structure of technical (8 runs) and biological replicates (3 animals). Specifically, it seems the 24 runs were treated as independent observations (given they are not independent, they should not be treated as such). Related to this, it would be important to also show results for each of the individual 3 subjects.

The statistical analyses are now described in more detail in the Methods section, and also pasted below. We now show individual maps in the supplementary data and refined our analysis by using paired-t-test for individual analyses (fixed-effect) and analysis of variance (three-factor ANOVA, mixed effect) to take into account that the data are not independent.

This has been modified in the Methods section as follows:

“The T-value maps for each monkey were then compared at the individual level via paired-sample t-tests and fixed effect analyses across the 8 runs for each marmoset (AFNI’s 3dttest++) and at the group level via a three-way analysis of variance and mixed effect analysis (AFNI’s 3danova3) using 8 runs per marmosets (for a total of 24 runs). To protect against false positives, a cluster-size correction derived from Monte Carlo simulations was applied to the resultant *t*-test maps for α=0.05 (using AFNI’s 3dClustsim).”

2. There was a general concern that the description of stimuli and the quantification of animal behavior was insufficient. This concern could be overcome by providing additional information and analyses, as listed below.2.1. L75/L158/L297 Please provide a more detailed description of the actions in each of the 'social' and 'nonsocial' videos in the Results and Methods sections, and provide the test videos as supplementary material.

Thank you, this is a good point. Additional details about the social and nonsocial videos have been added to the manuscript in two tables (Tables 1 and 2) and in the Results section as follows:

“In the social conditions, the actions performed in the videos included eating, playing, fighting and grooming between two marmosets. […] A more detailed description of these actions is provided in tables 1 and 2 in the Methods section.”

The videos are available upon request due to commercial copyright. Please see new statement in the manuscript.

2.2. L 93 "AFNI's 3dttest++" Please describe in the Methods section the kind of test that is coded by this AFNI function (fixed effect? Mixed effect analysis? corrected for multiple comparison?).

Details about this function have been added to the Methods section as follow:

“The T-value maps for each monkey were then compared at the individual level via paired-sample t-tests and fixed effect analyses across the 8 runs for each marmoset (AFNI’s 3dttest++) and at the group level via a three-way analysis of variance and mixed effect analysis (AFNI’s 3danova3) using 8 runs per marmosets (for a total of 24 runs). To protect against false positives, a cluster-size correction derived from Monte Carlo simulations was applied to the resultant *t*-test maps for α=0.05 (using AFNI’s 3dClustsim).”

2.3. Figure 1: What is the baseline used for comparison in the GLM and tests? Were all the baseline times concatenated? A baseline in the middle or at the end or at the beginning of each run?

The baseline used for comparison in the GLM and tests using AFNI corresponds to the regressors of no interest during the run (constant, drift, etc). Here it also included the no stimulation blocks and estimated motion parameters.

This has been added in the Methods section as follow:

“All four conditions were entered into the same model, along with polynomial (N = 5) detrending regressors, bandpass regressors, and the motion parameters derived from the volume registration described above (these nuisance regressors correspond to the baseline variance for the statistical analysis model).”

2.4. L192 "It should be noted that none of the oculomotor areas were activated by this contrast, precluding any confound with eye movements". This is an important information appearing only in the Discussion. Please describe in the Methods section how it was assessed, and in the Results section provide more details on this result.

This has been added to the manuscript as follows:

Methods section:

“The contrast between the social and nonsocial condition as well as the interaction effect [social vs. social scrambled] minus [nonsocial vs. nonsocial scrambled] were compared to previous fMRI and electrophysiological studies performed in marmosets (Ghahremani et al., 2017; Johnston et al., 2019; Schaeffer et al., 2019; Selvanayagam et al., 2019) to check if the observed activations may be related to eye movements.”

Results section:

“We did not observe any activation in area 8aV, indicating that he differences are not primarily related to saccades.”

Discussion section:

“Eye movements, and particularly visual saccades mostly activate the lateral intraparietal area (LIP) and the frontal eye field (FEF) in primates. […] The social condition may have elicited more small saccades related to the exploration of the facial expression and the interaction itself compared to the nonsocial interaction which may involve more large explorations.”

2.5. L279-280 "this camera and setup did not allow a proper recording of eye movements": Did it allow to monitor other interesting information: e.g. how long eyes were open during each condition, and/or to monitor blinks?

Unfortunately, the quality of the recordings was poor. We cannot assess how long the eyes were open during each condition as we sometimes lost the tracking signal. It means that even if the eyes were still open, the signal would be recorded as eyes close. Video-based eye tracking relies on tracking the center of the pupil which is extremely large in marmosets. Therefore, even small drops of the eye lids change the center of the pupil or lead to complete signal loss.

2.6. L296-306: Was there one video per block? Is there a difference in motion or luminance between conditions?

Two video clips were displayed within a block (6 seconds each) and 20 different videos clips were used in total. The videos within the same run were different between the two repetitions of the block condition.

Motion and luminance varied across videos, depending of the action and where the action took place, but we tried as much as possible to counterbalance these differences across the selected videos, conditions and order of presentation. In addition, to preserve motion and luminance components, the same random rotation matrix was used for each frame for the scrambled versions of the videos.

This has been added in the Methods section as follows:

“Two video clips were displayed within a block (6 seconds each) and 20 different videos clips were used in total. The videos within the same run were different between the two repetitions of the block condition but played following the same order. Tables 1 and two describe which action were performed for each social (Table 1) and nonsocial (Table 2) blocks.”

“To preserve motion and luminance components, the same random rotation matrix was used for each frame in the scrambled conditions.”

“As motion and luminance varied across videos, mainly due to where the action took place, we counterbalanced these differences across the selected videos, conditions and order of presentation to avoid any link with the condition block itself.”

2.7. L307-312: Are conditions presented in reverse order the second time during the run? Which baseline is used for contrasts presented in figure 1? Where the animals rewarded with juice during the scan? If so, for what type of behavior (fixation during baseline? looking at the videos, looking at the screen? Staying engaged?).

Conditions were presented in the same order the second time within the same run but this order was counterbalanced across runs. The baseline for the contrasts in Figure 1 corresponded to the regressors of no interest (no stimulation period, movement regressors, see point 2.3).

For this task, no reward was provided during the scan. Marmosets only received pudding before and after the scan session.

This has been added to the Methods section as follow:

“The videos within the same run were different between the two repetitions of the block condition but played following the same order.”

“No reward has been provided to the monkeys during the scan session.”

2.8. Additional information should be given regarding time courses of regions highlighted on maps and in the discussion, ie PG, PMv, AIP, 8A/45.

The PSC analysis includes now twelve regions of interest corresponding to the regions defined by the Paxinos atlas and based on the activation maps and regions highlighted in the discussion (areas 25, 10, 14c, VIP, PG, AIP. 6va, 6vb, 46d, 45, 8av and 8ad).

Figures, Results and Methods sections have been modified accordingly to this new data.

2.9. How many video clips were used in total? How many runs/sessions were collected?

Twenty different videos clips were used in total. Monkey 1 performed 8 runs in 4 sessions; Monkey 2, 11 runs in 5 sessions and Monkey 3, 9 runs in 4 sessions keep only 8. We selected 8 runs per monkey (based on their best engagement as assessed by the camera monitoring across each session) to have the same number across subjects and run the group analysis.

This has been added to the Methods section as follow:

“Monkey 1 performed eight runs in four sessions, Monkey 2 eleven runs in five sessions and Monkey 3 nine runs in four sessions. We selected eight runs per monkeys to have the same number of runs across subjects for the group analysis (the selection was based on the best engagement of the animal as assessed by visual monitoring during the sessions).”

3. Contrasts in Figure 2 and 3 are shown at p<0.05, uncorrected. Given the large number of voxels, whole brain tests should be corrected for multiple comparisons. Alternatively, the should be displayed at a more conservative uncorrected threshold (e.g., p<0.001 as for) as in Figure 1.

As mentioned previously, we have performed statistical analysis at both individual and group levels using paired t-tests and analysis of variance (three-factor ANOVA) but also the clustering method derived from Monte Carlo simulations to protect against false positives (AFNI’s 3dClustsim).

This has been modified in the Methods section as follow:

“The T-value maps for each monkey were then compared at the individual level via paired-sample t-tests and fixed effect analyses across the 8 runs for each marmoset (AFNI’s 3dttest++) and at the group level via a three-way analysis of variance and mixed effect analysis (AFNI’s 3danova3) using 8 runs per marmosets (for a total of 24 runs). To protect against false positives, a cluster-size correction derived from Monte Carlo simulations was applied to the resultant *t*-test maps for α=0.05 (using AFNI’s 3dClustsim).”

4. The authors isolate areas involved in observing social interactions by comparing responses to social and non-social interactions. To better control for low level visual features, please report the interaction effect ([social minus social scrambled] minus [nonsocial minus nonsocial scrambled]).

This contrast has been added and corresponds now to the new Figure 3 (and Figure 3—figure supplement 1 for individual maps).